# An omics-based framework for assessing the health risk of antimicrobial resistance genes

An-Ni Zhang[1,2], Jeffry M. Gaston [3], Chengzhen L. Dai [2], Shijie Zhao[2], Mathilde Poyet[2,4,5], Mathieu Groussin [2,4,5], Xiaole Yin[1], Li-Guan Li[1], Mark C. M. van Loosdrecht [6], Edward Topp[7], Michael R. Gillings [8], William P. Hanage[9], James M. Tiedje[10], Katya Moniz[2], Eric J. Alm[2,4,5] & Tong Zhang [1,11,12 ✉]

Antibiotic resistance genes (ARGs) are widespread among bacteria. However, not all ARGs pose serious threats to public health, highlighting the importance of identifying those that are high-risk. Here, we developed an 'omics-based' framework to evaluate ARG risk considering human-associated-enrichment, gene mobility, and host pathogenicity. Our framework classifies human-associated, mobile ARGs (3.6% of all ARGs) as the highest risk, which we further differentiate as 'current threats' (Rank I; 3%) - already present among pathogens - and 'future threats' (Rank II; 0.6%) - novel resistance emerging from non-pathogens. Our framework identified 73 'current threat' ARG families. Of these, 35 were among the 37 high-risk ARGs proposed by the World Health Organization and other literature; the remaining 38 were significantly enriched in hospital plasmids. By evaluating all pathogen genomes released since framework construction, we confirmed that ARGs that recently transferred into pathogens were significantly enriched in Rank II ('future threats'). Lastly, we applied the framework to gut microbiome genomes from fecal microbiota transplantation donors. We found that although ARGs were widespread (73% of genomes), only 8.9% of genomes contained high-risk ARGs. Our framework provides an easy-to-implement approach to identify current and future antimicrobial resistance threats, with potential clinical applications including reducing risk of microbiome-based interventions.

[1] Environmental Microbiome Engineering and Biotechnology Laboratory, Department of Civil Engineering, The University of Hong Kong, Hong Kong SAR, China. [2] Department of Biological Engineering, Massachusetts Institute of Technology, Cambridge, USA. [3] Google, Cambridge, USA. [4] Center for Microbiome Informatics and Therapeutics, Massachusetts Institute of Technology, Cambridge, USA. [5] The Broad Institute of MIT and Harvard, Cambridge, USA. [6] Department of Biotechnology, Delft University of Technology, Delft, The Netherlands. [7] London Research and Development Centre (LRDC), Agriculture and Agri-Food Canada, London, ON, Canada. [8] Department of Biological Sciences, Macquarie University, Sydney, NSW, Australia. [9] Center for Communicable Disease Dynamics, Department of Epidemiology, Harvard TH Chan School of Public Health, Boston, USA. [10] Department of Plant, Soil and Microbial Sciences and of Microbiology and Molecular Genetics, Michigan State University, East Lansing, MI, USA. [11] School of Public Health, The University of Hong Kong, Hong Kong SAR, China. [12] Center for Environmental Engineering Research, The University of Hong Kong, Hong Kong SAR, China. ✉email: zhangt@hku.hk

Antimicrobial resistance (AMR) has been declared a global public health threat by the U.S. Center for Disease Control (CDC)[1] and World Health Organization (WHO)[2]. Mitigating this threat requires a multipronged approach that considers the risks of novel AMR emergence and transfer to pathogens, transmission of resistant pathogens, and targeted, evidence-based strategies for reducing the factors that contribute to each of those steps. Here, we focus on identifying those antibiotic resistance genes (ARGs) that have significant potential to endanger public health.

However, identifying which ARGs pose a threat to human health is not straightforward. Genes that are believed to confer antibiotic resistance—including those predicted by sequence homology—are ubiquitous among bacteria, fulfilling numerous biological roles such as efflux systems[3] and cell–cell signaling[4]. Only a small fraction of these genes pose a threat to human health[5]. Thus, identifying "high-risk" resistance genes among the many thousands of presumptive ARGs is a critical step to tackle this global problem with cost-effective approaches.

ARG risk to human health varies according to a number of factors, including host pathogenicity, genetic context, and likelihood of transfer to human pathogens. Intrinsic resistance to colistin (a last-resort antibiotic) was identified decades ago[6], but the genes' low potential for horizontal gene transfer (HGT) limited their spread and clinical impact. By contrast, the mobilized colistin resistance gene, mcr-1, has spread rapidly into seven pathogenic species across 31 countries, largely driven by plasmid-mediated HGT[7], and has been reported to be highly prevalent in human and livestock fecal samples[8,9]. Gene mobility and host pathogenicity have been proposed as evaluative factors for ARG risk by previous studies[10–13], and were also included in this study.

Previous frameworks for assessing ARG risk to human health remain theoretical and unimplemented due to the limited availability of clinical and experimental data[10,13]. Here, we apply a microbial ecology approach to design and implement a practical risk ranking framework.

Our risk framework is a decision tree based on the assessment of three criteria: (1) enrichment in human-associated environments, (2) gene mobility, and (3) presence/absence in ESKAPE pathogens (host pathogenicity). This simple framework yields four risk categories. ARGs that do not meet the first criterion are assigned to Rank IV (lowest risk); those that meet the first, but not the second, are assigned to Rank III; those that meet the first and second but not the third, Rank II; and those that meet all three criteria, Rank I (the highest risk).

The first metric in our framework considers the fact that ARGs that are much more abundant in anthropogenically impacted environments than in non-impacted environments are most likely either to be associated with human or livestock microbiomes, or to be directly selected for resistance to clinical or livestock antibiotics, or both. We refer to this as the enrichment of putative ARGs in "human-associated" environments. Conveniently, ARG enrichment in human-associated environments can be assessed using environmental metagenomic data, allowing easy classification of ARGs that are less likely to be clinically relevant as low risk.

While the three criteria we use could arguably be applied in a different order, the framework described above results in a sensible risk hierarchy: ARGs that are not human-associated (Rank IV) are the least likely to endanger human health. Among human-associated ARGs, non-mobile ARGs (Rank III) are less likely to contribute to an emergence of new resistance in pathogens. This leaves mobile ARGs that pose the highest risk of contributing to new or multidrug resistance in pathogens, in the future (Rank II—not yet present in pathogens) and at present (Rank I—already present in pathogens).

We applied the list of Rank I–II ARGs to 1,921 bacterial genomes from healthy stool donors, as a demonstration of how this approach can be used to screen candidate strains for microbiome-based therapeutics for high-risk ARGs. This framework takes a microbial ecology approach to address a difficult, clinically relevant question, and provides new insights on potential approaches beyond microbiome-based therapeutics.

## Results

**Less than 30% ARGs are associated with clinical antibiotics.** We performed a literature survey on the concentrations of clinical and livestock antibiotics across environments. Total environmental concentration of antibiotics varied by two orders of magnitude, with the lowest concentrations in pristine natural environments such as permafrost and marine sediment, and the highest concentrations being in strongly human-associated environments such as wastewater treatment plants (WWTPs) and industrial livestock waste streams (Fig. 1a, references in Supplementary Data 1, methods in Supplementary Methods). Clinical and livestock antibiotics are widely present and highly abundant in human-associated environments, which strongly select for clinically relevant ARGs[14] in those environments.

In contrast, the total abundance of all ARGs shows no significant difference (<10-fold difference and $p = 0.2$ by ks-test) in their overall abundance across these environments, as reported by a previous metagenomic survey[15] (Fig. 1b and Supplementary Fig. 1a). However, investigating the composition of ARGs in 854 metagenomes reveals that distinct groups of ARGs dominated in different environments, varying along a primary axis of anthropogenic (Fig. 1c). Thus, environmental concentrations of clinically relevant antibiotics were related to a subset of ARGs, which were not identified by the traditionally used risk metrics of mobility and host pathogenicity (Supplementary Fig. 2). To distinguish the subset of ARGs that correlated with anthropogenic impact, we defined "human-associated" ARGs as those being ≥100-fold more abundant in metagenomes of anthropogenically impacted environments than in metagenomes of non-anthropogenically impacted environments (Fig. 1b and Supplementary Fig. 3).

We found that the majority of ARGs were not "human-associated". We applied the risk framework to a total of 4050 ARGs from the Structured ARG Database[16] (referred to as ARGs of the initial set). We were able to assess 2579 ARGs of the initial set; the other 1471 remain unassessed because they were undetected in the metagenome dataset[15]. We found that 70% (1816 of 2579) of ARGs of the initial set were not "human-associated"; and thus, were assigned to Rank IV, the lowest risk category (Fig. 2b). Rank IV ARGs also show other low risk characteristics: the majority (83%, 1505 of 1816) of them were not found on any mobile genetic elements (MGEs) (Supplementary Fig. 4), and they were 5–10 times more abundant in non-anthropogenically impacted environments.

**Identification of 3.6% of ARGs as current and future threats.** We found that the majority of human-associated ARGs were not carried by any mobile genetic element (MGEs). Of all human-associated ARGs, 81% (618) of them were not detected on any MGEs (plasmids, integrons, or in the intestinal microbiome mobile element database[17,18]) and were therefore classified as Rank III (Fig. 2b).

Of all human-associated and mobile ARGs, the majority (84%, 122) of them were already present in ESKAPE pathogens (i.e., Enterococcus faecium, Staphylococcus aureus, Klebsiella pneumoniae, Acinetobacter baumannii, Pseudomonas aeruginosa, and Enterobacter species), and thus, were classified as Rank I. The remaining 16% (23 genes) that were not present in any ESKAPE

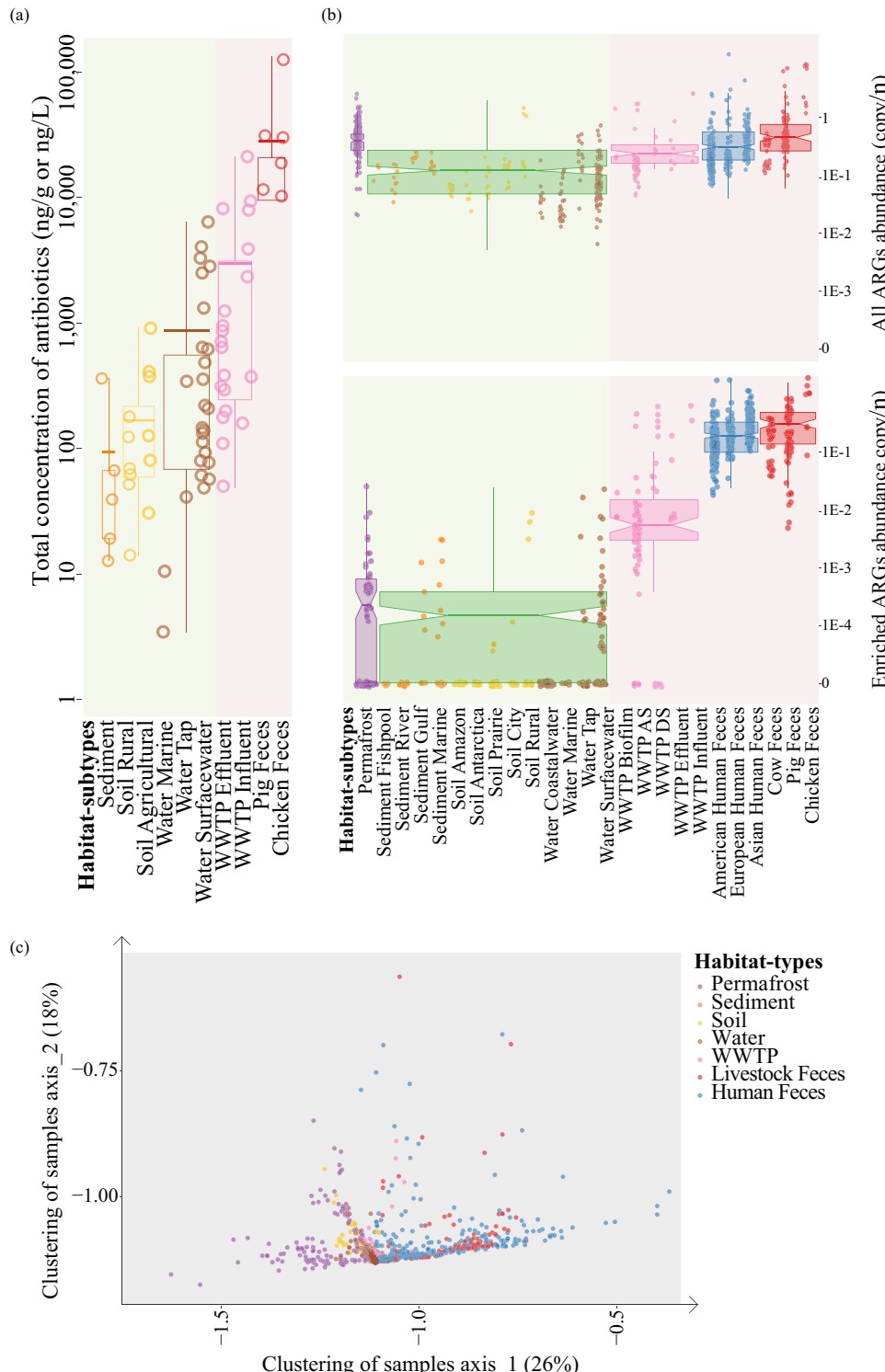

**Fig. 1 Survey into the distribution of total antibiotics and antibiotic resistance genes (ARGs) in diverse environments to inform the development of the risk framework. a** The total concentrations of total antibiotics (filled circles for ng/L and open circles for ng/g) in different habitats from 30 studies (Supplementary Data 1). Each circle represents a sample, with color representing its habitat. **b** The abundance of total ARGs and abundance of ARGs in different habitats. Each circle represents a metagenomic sample, with color representing its habitat. The abundance of ARGs was normalized into copy of ARGs per cell (ARGs-OAP, more details in "Methods" section). **c** PCA plotting ARG composition clustered 854 metagenomic samples from seven global eco-habitats into undisturbed natural environments (green panel covering permafrost, soil, sediment, and water) and human associated environments (red panel covering wastewater treatment plants (WWTPs), animal feces and human feces). Samples were clustered into three habitats (from undisturbed natural habitats, to WWTPs, to feces) along the primary axis of "human-association", with its color representing its habitat. Data in **a** are presented as mean values (center) and 25%, 75% percentiles (bounds of box). Data in **b** and **c** are presented as median values (center) and 25%, 75% percentiles (bounds of box). The minima and maxima represent the range of the data.

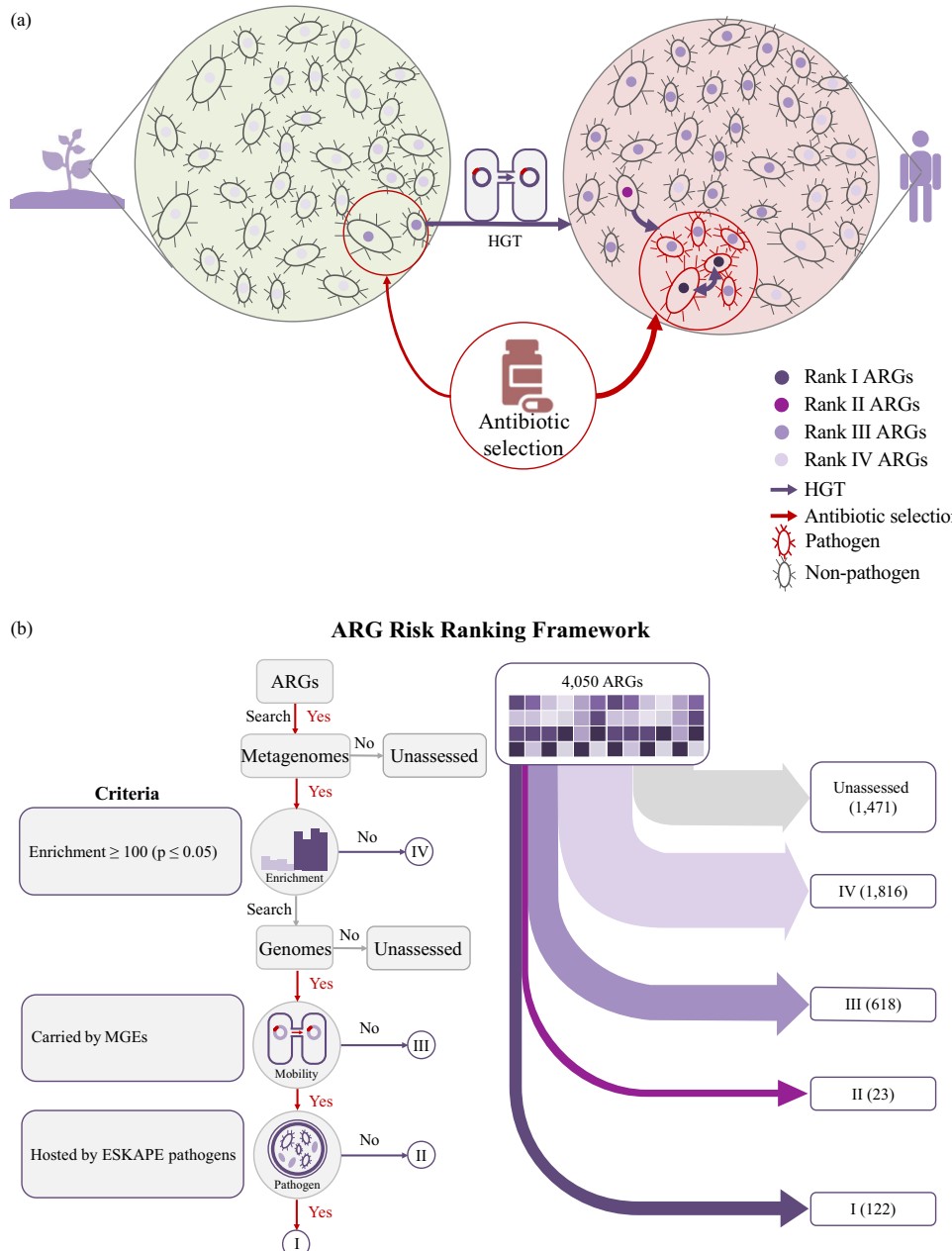

**Fig. 2 A conceptual model of the evolution and emergence of antibiotic resistance genes (ARGs) and a framework for assessing ARG risk to human health. a** We designed a conceptual model to demonstrate the evolution and emergence of ARGs accelerated by selective agents of antibiotics. **b** Based on this model, we further designed a framework to reflect the natural progression of the evolution and emergence of ARGs and the risk ranks of all ARGs reference sequences of the initial set in the Structured ARG Database v1.0. The color represents the rank of ARGs (Ranks I–IV). The relevant data was obtained by searching the ARGs in all available bacterial genomes and plasmids from NCBI, mobile genetic element (MGE) databases, and 854 global metagenomes using the ARGs Online Searching Platform. HGT horizontal gene transfer, ARGs antibiotic resistance genes.

pathogen were classified as Rank II (Figs. 2b and 3, Supplementary Data 2–4).

We designated Rank I ARGs as "current threats" that have the highest potential to contribute to multidrug resistance in pathogens via multiple dangerous characteristics: a wide host range facilitated by mobility (94%, i.e., 116 were shared across species, and 76%, i.e., 93 across genera) and a wide niche adaptation (75%, i.e., 92 were carried by both pathogens and non-pathogens). We propose that Rank II ARGs, especially 15 genes without Rank I homologs (Fig. 3), represent future threats that could transfer to pathogens as new forms of resistance[10,19] because Rank II ARGs were found in abundant gut commensals

(e.g., *Lactobacillus* and *Bacteroides* species) or close relatives to pathogens.

**ARGs in the same gene family exhibit divergent risks**. Our framework assessed ARG risk at the level of gene sequence rather than gene family. We found that homologous ARGs in the same gene family can pose substantially different risks because they displayed divergent characteristics, such as host range, mobility potential, and ecological distribution. Homologs of the same gene family exhibited divergent "host ranges", varying from 1 to 30 bacterial genera, with most (84%, 2164) ARG sequences of the initial set being carried by only 1–2 bacterial genera, but several

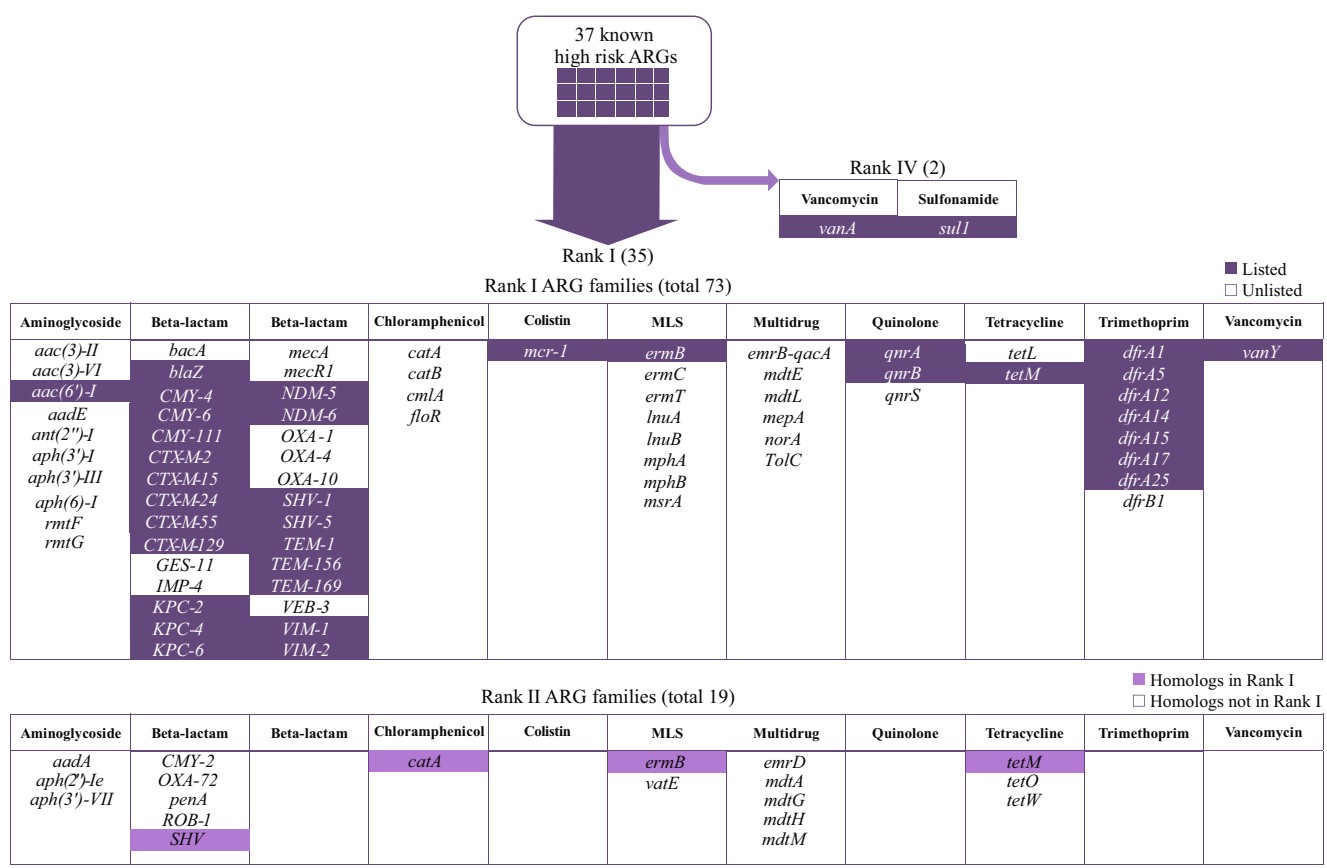

**Fig. 3 The risk ranks of 37 known high risk ARG families identified by World Health Organization (WHO) and literature review (highlighted in dark purple) (referred to as "listed" ARGs), 73 Rank I ARG families of the initial set, and 19 Rank II ARG families of the initial set.** "Listed" (WHO-listed) represented Rank I ARGs that were reported to cause problems in hospitals and/or have been widespread on mobile genetic elements by previous literatures (highlighted in dark purple). "Unlisted" represented Rank I ARG families that were not reported in previous studies as high risk ARGs. "Homologs in Rank I" represented Rank II ARGs with homologous Rank I ARGs of the same gene family (highlighted in light purple). ARGs antibiotic resistance genes.

(2%, 51) being distributed across >10 genera. Conversely, we found that while carrying more than one homolog of a high-risk ARG was rare among bacteria (0.14% 42 of 29,595 instances), carrying multiple homologs of low-risk ARGs from the same gene family was widespread (19%, 23,117 of 121,610 instances). This observation is consistent with low-risk ARGs conferring different ecological functions than high-risk ARGs. Lastly, we found that the same reference ARGs of the initial set displayed distinct sequence variants in different habitats, with far fewer variants (mostly 100% amino-acid similarity) in feces than in WWTPs or in non-anthropogenically impacted environments (mostly lower than 85% amino-acid similarity) (Supplementary Fig. 5). Thus, it appears likely that environmental ARGs could confer different phenotypic properties than those of ARGs in feces, or other known ARGs that are annotated as the same gene family. This highlights the importance of conducting ARG research and surveillance at the sequence-level in future studies.

**Agreement between Rank I and known high-risk ARGs.** This simple framework largely agreed with assessments of ARGs by human experts. We compared our Rank I ARGs against a list of 37 ARG families of high clinical concern that have been reported by WHO[20] (referred to as "WHO-listed Rank I ARGs") and other literature (see Supplementary Methods) to have caused antibiotic treatment failure in hospitals across the world and/or to be widespread on MGEs (Fig. 3). Since the publicly available data was primarily at the gene family level rather than sequence level, we first clustered our 122 Rank I ARG sequences into 73 Rank I

ARG families before comparison. Of the 37 "WHO-listed" Rank I ARG families, our framework successfully identified 35 of them as Rank I. The remaining two (*vanA* and *sul1*, which confer resistance to vancomycin and sulfonamide, respectively) met the requirements of gene mobility and host pathogenicity, but no ARGs in these two families showed a significant enrichment in human-associated environments, and were therefore categorized as Rank IV (see Discussion). In addition, the framework identified 38 Rank I ARG families that have not yet been reported as high risk (referred to as "unlisted" Rank I ARGs), but have shown a strong clinical relevance similar to those known high-risk ARG families as supported by this study (Supplementary Fig. 6c) and other studies (such as *IMP-4* and *OXA*[21,22]). We propose that these 38 "unlisted" Rank I ARG families do represent a current threat but are largely under-reported due to limited genotypic assays and genomic analysis in the clinical setting.

**Validation of Rank I-II ARGs by the recent clinical datasets.** To validate the performance of our framework, we evaluated whether there was evidence that Rank I ARGs not already on the WHO list and Rank II ARGs were, in fact, high risk, using clinical data from the two years following the training datasets. Specifically, we evaluated which ARGs were the most prevalent among the riskiest pathogens (i.e., hospital pathogens[23]) in the more recent dataset, and which ARGs were most likely to become newly present in pathogens in those 2 years. We analyzed three datasets collected from 02/20/2019 and 02/10/2021 (Fig. 4a, see "Methods" section), whose data were not included in the training

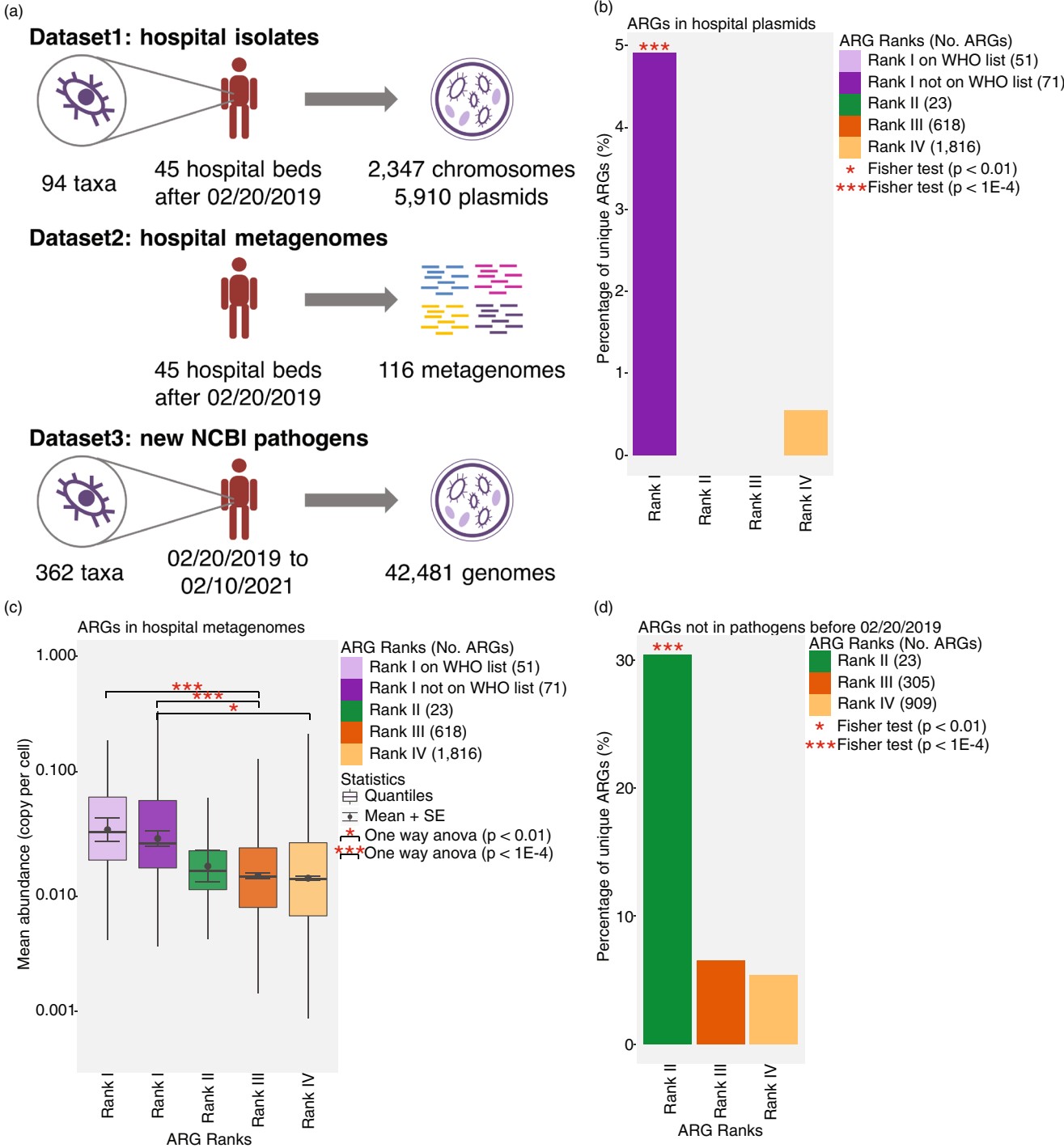

**Fig. 4 Validation of the framework in this study by three clinical datasets collected after the initial analysis between 02/20/2019 to 02/10/2021 as the training datasets were collected on 02/19/2019. a** The content of three datasets. Dataset1 contains the chromosomal genomes and plasmids collected from opportunistic pathogen isolates cultured from a tertiary hospital after 02/20/2019. Dataset1 covers a total of 94 taxa (2,347 chromosomal genomes and 5,910 plasmids). Dataset2 contains 116 metagenomes collected after 02/20/2019 from a tertiary hospital. Dataset3 contains all NCBI pathogen genomes submitted in the last 2 years (02/20/2019 to 02/10/2021). Dataset3 covers a total of 362 taxa (42,481 genomes). ARG sequences of the initial set were searched in three testing datasets using the same criteria described before (90% aa similarity over 80% aa hit length). **b** the percentage of unique ARGs of different Ranks in hospital plasmids. Asterisk of box represent the $p$ value for fisher exact test. $n = 16$ (No. of unique ARGs). **c** The mean abundance of ARGs of different Ranks in 116 hospital metagenomes. Quantiles (represented as bars), mean (represented as points), and standard error (represented as error bars) were computed for all ARGs in an ARG Rank. Asterisk of box represent the $p$ value for one way anova test. $n = 1393$ (No. of unique ARGs detected in hospital metagenomes). Adjustments were made for multiple comparisons. Data are presented as mean values ± SE (error bars), median values (center of box), and 25%, 75% percentiles (bounds of box). The minima and maxima represent the range of the data. **d** The percentage of unique ARGs that were found in pathogens after 02/20/2019 and were not detected in pathogens before 02/20/2019. Asterisk of box represent the $p$ value for fisher exact test. $n = 76$ (No. of unique ARGs newly present in pathogens). Adjustments were made for multiple comparisons. The color represents the rank of ARGs (Rank I on WHO list, Rank I not on WHO list, Rank II-IV). ARGs antibiotic resistance genes.

datasets (collected before 02/20/2019) that were used to develop the framework. We searched ARG sequences of the initial set in three testing datasets using the same criteria described before (90% aa similarity over 80% aa hit length).

To evaluate the importance of our Rank I ARG classification, we split our Rank I ARG group into two subgroups: "WHO-listed" Rank I ARGs, which were identified by the WHO and previous literature as being highly clinically relevant, and "unlisted" Rank I ARGs which were not. We expect that higher-risk ARGs will be more commonly found in opportunistic pathogens in hospital isolates and especially on their plasmids[23] and will be more abundant in hospital metagenomes, and we evaluated the probabilities of two categories of Rank I ARGs to be found in these locations. Dataset1 consisted of chromosomal genomes and plasmids from opportunistic pathogen hospital isolates[24], and included 94 taxa (2,347 genomes and 5,910 plasmids). We found that ARGs on hospital plasmids were significantly enriched in Rank I ARGs (Fisher exact test $p < 1E-4$) (Fig. 4b), and that all of these were "unlisted" Rank I ARGs. Rank I ARGs were not enriched in hospital chromosomes (Fisher exact test $p > 0.05$). Dataset2 consisted of 116 metagenomes collected from the same hospital[24]. ARG abundance in these metagenomes was the highest for Rank I, and decreased for each lower risk category (Fig. 4c). The abundance of "unlisted" Rank I ARGs was comparable to that of "WHO-listed" Rank I ARGs (one way anova $p > 0.05$), and was significantly higher than that of both Rank III and Rank IV ARGs (one way anova $p < 1E-4$ and $p < 0.01$). The higher prevalence of "unlisted" Rank I ARGs in recent hospital pathogen plasmids and their abundance in hospital metagenomes is consistent with a high risk profile.

Our framework proposes that Rank II ARGs should be considered "future threats": ARGs that are not yet present in, but are at high risk of transferring into, human pathogens. To validate this hypothesis, we evaluated ARGs that were not previously present in pathogens (i.e., before 02/20/2019), and determined which were newly present in pathogens in a more recent dataset (i.e., after 02/20/2019). This dataset (Dataset3) consisted of all NCBI pathogen genomes submitted in the last two years, and included 362 taxa and 42,481 genomes[25]. Rank I ARGs, by definition, were previously found in pathogens before 02/20/2019 and therefore were not included in this analysis. We found that ARGs detected in pathogens in Dataset3 were significantly enriched in Rank II ARGs (Fisher exact test $p < 1E-4$) (Fig. 4d). Moreover, even though Rank II comprises only 23 ARGs, Rank II ARGs transferred into pathogens at a higher rate (5×) (30.4%, 7 of 23) than Rank III (6.6%, 20 of 305) and Rank IV ARGs (5.4%, 49 of 909), suggesting that they are at elevated risk of transferring into pathogens. Based on the more recent dataset, the Rank II ARGs that were newly found in pathogens should now be classified as Rank I ARGs. These include two beta-lactam resistance genes (SHV and OXA-72), one aminoglycoside resistance gene (aph(2")-Ie), one multidrug resistance gene (emrD), two MLS resistance genes (vatE and ermB), and one chloramphenicol resistance gene (catA) (Supplementary Data 5).

## Discussion

Mitigating the global threat of antimicrobial resistance requires a multi-pronged approach that considers the risks of novel antibiotic resistance emergence and transfer to pathogens, transmission of resistant pathogens, and targeted, evidence-based strategies for reducing the factors that contribute to each of those steps (Supplementary Fig. 7). In this study, we focused on identifying "high-risk" ARGs that may pose significant threat for human health. We propose a framework by which different ARGs may be classified into different risk categories on the basis of their

potentials to contribute to the emergence of new or multidrug resistance in pathogens. The framework is an easy-to-implement decision tree that uses the factors of human-associated enrichment, gene mobility, and host pathogenicity.

ARGs in non-pathogens may provide a reservoir of antimicrobial resistance for pathogens. We found that 84% (122) of mobile, enriched ARGs of the initial set were already present in pathogens and many of them (75%, 92) were shared across non-pathogens and pathogens. This suggests that most high-risk ARGs (Rank I) were either transmitted from non-pathogens (Rank II) into pathogens, or originated from pathogens (Rank III) (Fig. 2a). The possibility that a Rank I ARG could originate from non-pathogens highlights the importance to design a risk category (Rank II) representing the potential for emerging resistance from transfer between non-pathogens to pathogens[10,26]. Future studies into the origin and evolution of Rank I ARGs can provide better insights into the factors that contribute to the emergence of high-risk ARGs and provide guidance into proactive strategies to reduce those factors and mitigate this threat. For example, ARG phylogenetic analysis can help to differentiate horizontally acquired resistance from mutation-based resistance, which evolves differently and therefore requires different strategies to control. The former one would more likely be driven by anthropogenic pollution with selective agents and microbes of human or domestic animal origin; and the latter would more likely be selected by antibiotic treatment inside human, domestic animal body[10], or in the biological reactors treating high-strength antibiotic-containing wastewater.

Of 37 previously identified "WHO-listed" Rank I ARG families, our framework identified two families (vanA and sul1) as Rank IV, because they were highly prevalent in "non-human-associated" environments such as soil and water. However, the environments in which sul1 sequences showed a 100-fold higher abundance were in fact environments with a high likelihood of anthropogenic contamination (e.g., surface water and agricultural soil) (Supplementary Fig. 6a, b). Because sulfonamide is one of the earliest and most widely applied antibiotics since 1930s, we considered that sul1 has already been widespread in many non-human-associated environments because of anthropogenic contamination.

The vanA sequences, however, were 2–53-fold more abundant in non-human-associated environments than human-associated environments. The dominant mobile vanA sequence carried by ESKAPE pathogens (NP_878016) was 10–100-fold more abundant in permafrost and soil samples than in human feces. It indicates that vanA in human microbiome could have originated from natural microbes thousands of years before the emergence of industrialized antibiotics[27]. However, the most dominant and widespread vanA sequence (KF478993.1.gene3.p01) was not found in any pathogen or on any MGE, and shared 62% amino-acid similarity to the mobile vanA (NP_878016). This means sequence-level analysis for vanA is essential for future studies and surveillance. Because vancomycin is an antibiotic in the "Watch" category of WHO AWaRe classification, its use as the first and second choice treatment is limited[28].

To take into consideration the clinical importance for different classes of antibiotics, such as (a) the optimal usage of antibiotics, and (b) potential for antimicrobial resistance[29], we assigned the antibiotics targeted by Rank I-II ARGs to the corresponding stewardship group ("Access", "Watch", and "Reserve") according to the WHO AWaRe classification[29]. The antibiotics targeted by Rank I-II ARGs covered 71 of 180 antibiotics listed by WHO AWaRe, in which 21.1% (15 of 71) are "Access" antibiotics (with a wide target range of susceptible pathogens and lower antimicrobial resistance potential), 71.8% (51 of 71) are "Watch" antibiotics (with the highest priority among the Critically

Important Antimicrobials for Human Medicine and higher resistance potential), and 7.0% (5 of 71) are "Reserve" antibiotics (which should be reserved for treatment of infections due to multidrug-resistant organisms). Currently most (77.3%, 17 of 22) "Reserve" antibiotics, which are also called "last resort" antibiotics, have no high-risk resistance potential (Rank I-II ARGs), except for Plazomicin, Eravacycline, Minocycline (IV), Omadacycline, and Dalfopristin-quinupristin. However, antibiotic targets of Rank I-II ARGs had significantly more "Watch" antibiotics (Fisher exact test, $p < 0.05$) than the antibiotics that are not targets of Rank I-II ARGs. Because "Watch" antibiotics are prioritized by WHO AWaRe as key targets of stewardship programs and monitoring, the surveillance and control of the transmission of bacterial strains carrying Rank I-II ARGs to human should be the first step[10].

One way to control the transmission of high-risk AMR to humans is to screen out high-risk strains for microbiome-based therapeutic and live biotherapeutic products, which is minimally regulated across the world[30–34] (Supplementary discussion). To demonstrate this, we surveyed antibiotic resistance of 1,921 genomes of representative human gut microbiome strains cultured from 59 healthy donors[35,36], who had not consumed antibiotics in the 6 months prior to sample collection (Supplementary Data 6 and 7). We collected data from both fecal microbiota transplantation (FMT) donors (489 strains) and non-FMT donors (1,432 strains). Samples from FMT donors were collected from Broad Institute-OpenBiome Microbiome Library (BIO-ML), and all donors were screened for antibiotic resistant bacteria including ESBL-producing organisms, carbapenem-resistant *Enterobacteriaceae* (CRE), methicillin-resistant *Staphylococcus aureus* (MRSA), and vancomycin-resistant *Enterococci* (VRE). As Rank I ARGs, by definition, would be highly prevalent in pathogens, we classified all strains into 227 pathogenic strains exhibit multidrug resistance and virulence (ESKAPE and multidrug resistant *Escherichia coli*) and 1,694 non-pathogenic strains, covering a total of 385 non-pathogenic species (Supplementary Fig. 8). Since we observed no significant difference between genomes from FMT donors and non-FMT donors (Supplementary Fig. 9) in terms of high-risk ARG prevalence in non-pathogenic strains ($p = 0.89$ and 0.99 for Rank I–II, respectively by ks test) and that in pathogenic strains ($p = 0.64$, 1.00 for Rank I–II, respectively by ks test), we combined the results of all donors for the following discussion.

We found that ARGs were ubiquitous among human gut commensal species but high-risk ARGs were mostly enriched in pathogenic strains. We detected ARGs in 72.6% (1229 of 1694) of all non-pathogenic strains, covering a broad lineage of commensal species including *Bacteroides*, *Bifidobacterium*, and *Lactobacillus* (Supplementary Data 6 and 7 and Supplementary Fig. 8). How many of these species pose a serious threat to human health? If we exclude all these bacterial strains from microbiome-based therapeutics, we would remove the entire commensal species or genera that have unreplaceable functions in human gut because of a few non-mobile/intrinsic ARGs, e.g., *Lactobacillus ruminis* for an intrinsic bacitracin resistance gene (*bacA*) and *Bacteroides* species for intrinsic macrolide-lincosamide-streptogramin resistance genes (*mefA*, *ermG*) and intrinsic beta-lactamases (*CfxA2*, *CfxA3*). However, if we focused on the list of high-risk ARGs, only 150 non-pathogenic strains (8.9%) were found to carry Rank I-II ARGs (Supplementary Fig. 9). These non-pathogenic strains are more likely to contribute to new resistance in pathogens, and thus should be excluded from microbiome-based therapeutic. Moreover, we found that pathogenic strains pose a higher risk of known antibiotic resistance than non-pathogenic strains. We observed that high-risk ARGs were significantly enriched in pathogenic strains ($p = 2E-88$ by

Fisher exact test) and that most (69.9% 158 of 227) pathogenic strains were found to carry high-risk ARGs. Thus, we recommend excluding the entire set of pathogenic strains and the subset of non-pathogenic strains with high-risk ARGs from microbiome-based therapeutic applications.

Moreover, we found that the list of non-pathogenic strains that carried Rank I-II ARGs varied from donor to donor. These high-risk non-pathogenic strains covered a wide lineage of 88 bacterial species, including many commensal species (i.e., *Bacteroides* and *Bifidobacterium* species), while the majority of these species (79%, 565 of 715 bacterial strains) did not carry any Rank I–II. This suggests that PCR assays or whole genome sequencing should be used to screen out high-risk bacterial strains, rather than a fixed species list. Therefore, to help control the transmission of high-risk ARGs, we propose that world-wide health authorities should require PCR assays or sequencing survey, as a standard safety evaluation for microbiome-based therapeutic usage and live biotherapeutic products.

The list of Rank I–II ARGs identified in this study can provide recommendations for regulating AMR through FMT in the future. FMT has been proven a promising treatment for some dysbioses of the human gut microbiome, e.g., *C. difficile* infection[37], inflammatory bowel disease (IBD)[38], and hematopoietic cell transplantation (HCT)[39]. It is a life-saving treatment with no alternative currently available in other ways. However, a potential risk of FMT is the transfer of multidrug resistant organisms[40] and/or high-risk ARGs that have the potential to contribute to new resistance in pathogens. Currently, antimicrobial resistant pathogens in FMT are strictly regulated by examining multidrug resistant organisms (MDRO) in donor FMT material, for example ESBL-producing *Enterobacteriaceae*, VRE, CRE, and MRSA requited by the Food and Drug Administration (FDA)[41]. Incorporating the risk framework proposed in this study with MDRO testing would help improve the safety of FMT. Applying the risk framework to FMT donor samples would further diminish the risk of novel resistance emerging from non-pathogens via ranking FMT samples by their abundance of Rank I–II ARGs in their gut microbiome metagenomes, and prioritizing FMT samples according to ARGs risk level. Moreover, this framework can be used in the clinical trial design to investigate which microbiota-based intervention results in the least burden of high risk ARGs. However, this idea requires further validation before implementation. First, the computational prediction of ARGs based on genotype does not always reflect the phenotypes. Future studies should investigate the efficiency of transferring high-risk ARGs through FMT using the metagenomes from FMT donors and recipients, and study the evolution and transmission of high-risk ARGs within person by collecting long-term follow-up metagenomes from FMT recipients. Moreover, only considering high-risk resistome is not sufficient for the assessment of the safety of FMT donors. Further investigation into the virulence factors, endotoxic chemicals, virome composition, and even debris of dead bacteria will help strengthen the guidance for the safety of FMT and other microbiome-based therapeutics.

As we move into an era of heightened molecular surveillance, it's important to interpret the risk of ARGs in environments rather than simply document their presence and concentration. This study provides an easy-to-implement framework and a bioinformatic tool to assess ARGs in genomes and metagenomes (details in "Methods" section), and offers insights for future studies to design and improve the screening guidelines and regulatory framework for FMT and microbiome-based therapeutics. The microbial ecology approach used in this framework also provides new insights on potential approaches beyond FMT. For example, this framework could be applied to prioritize bacterial strains carrying ARGs with high risk of environmental

dissemination and identify potential environmental hotspots by investigating sequencing data. Future studies can provide valuable applications to effectively preventing the emergence, and the transmission of ARGs into human pathogens by quantifying the risk of ARGs in environmental metagenomes using our framework. Specifically, for Rank I–II ARGs, time-effective and cost-effective molecular methods such as Nanopore and qPCR should be designed for fast detection and supervision[42]. Future studies and organizations could design new PCR primer sets targeting the list of Rank I–II ARGs identified in this study for real-time surveillance and for healthcare providers to deliver an effective antibiotic therapy[42]. One critical factor to consider when screening for ideal primers is the capability of amplifying, with a high degree of specificity, only ARGs of interest in a wide range of bacterial species. Moreover, applying ARG primers to the plasmidome and combining ARG primers with primers targeting MGEs (e.g., integrons[43]) via epicPCR[44] could be useful to target mobile high-risk ARGs. Additionally, risks of Rank III ARGs carried by ESKAPE pathogens could be mitigated by genomic analysis to improve antibiotic prescription for an effective treatment. Finally, the ongoing COVID-19 pandemic is likely to lead to a global shift in the volume of antibiotic use, in particular among those commonly used in respiratory bacterial superinfections. The consequences for the global distribution of ARGs remains to be seen.

This framework was limited by its lack of phenotypic factors, which are not currently available for most ARG sequences. Future studies could benefit from characterizing the phenotypes of ARGs in primary human pathogens[45] and human gut commensals, and provide clinical evidence. Additionally, the first criterion, enrichment in human-associated environments, was evaluated using datasets with limited metadata describing the level of anthropogenic impact, which had to be inferred. We controlled for the fecal contamination in the non-human-associated samples (Supplementary Fig. 10), but information on antibiotic usage is not readily available in the original publications. In this study, we only focused on ESKAPE pathogens. Future work would also benefit from comprehensive pathogen lists[46], and manual curation is highly recommended to differentiate commensals and opportunistic pathogens (i.e., *Bacteroides* and *Lactobacillus* species). Furthermore, we understand there could be other prioritization approaches which have focuses different from that of this study.

## Methods

**ARG identification in genomes and metagenomes**. Briefly, the ARGs Online Searching Platform[15] provided the presence and abundance of ARGs of the initial set in 54,718 NCBI bacterial genomes (downloaded on 02/19/2019) (after a screened by ≥50% completeness, <10% contamination, and curated by Genome Taxonomy Database[47]), 15,738 (all available after quality screening) NCBI plasmids (downloaded on 02/19/2019), and 854 global metagenomes of Illumina shotgun sequencing (downloaded on 02/19/2019). We further searched ARGs of the initial set in other MGEs databases (integrons[17] and intestinal microbiome mobile element database ImmeDB[18]). The search cutoff was set for genomes and MGEs as $e$-value of 1e−5, 90% aa similarity over 80% aa hit length; and for metagenomes as $e$-value of 1e−7, 80% aa similarity over 75% aa hit length[16,48–50]. The bias from sequencing depth and bacterial DNA ratio across samples was controlled by normalizing the copies of ARGs by the total number of bacterial cells (Supplementary Fig. 11).

**FMT datasets collected for the application of the framework**. We collected 1,921 representative human gut microbiome genomes[35,36] cultured from 59 healthy donors, 489 bacterial genomes isolated from the stool samples of 11 healthy FMT donors[35], and 563 human gut microbiome metagenomes from 84 FMT donors[35]. All human donors were healthy individuals with no antibiotic consumption from 6 months before sampling. The metagenomes consisted of 402 metagenomes of four donors with intensive sampling (206 samples over 536 days, 74 samples over 375 days, 59 samples over 201 days, and 63 samples over 144 days) and 161

metagenomes for 80 donors with sparse sampling (1–3 samples per individual over 2–460 days) (Supplementary Data 8).

**Clinical datasets collected for the validation of the framework**. To validate our framework, we analyzed three testing datasets collected from 02/20/2019 and 02/10/2021, whose data were not included in the training datasets collected before 02/20/2019. Dataset1 contains the chromosomal genomes and plasmids from opportunistic pathogen isolates cultured from a tertiary hospital, in which samples were collected from 179 sites associated with 45 hospital beds over 1.5 years (downloaded from https://t.co/bdZxADGM7z)[24]. Dataset1 covers a total of 94 taxa (2,347 genomes and 5,910 plasmids) (Supplementary Data 9 and 10). Dataset2 contains all 116 metagenomes collected from the same study[24] of Dataset1 (by platform Illumina HiSeq 2000, accession numbers ERX3237365-ERX3237728, ERX3667056-ERX3667128, ERX3669272-ERX3669296) downloaded from https://www.ebi.ac.uk/ena/browser/view/PRJEB31632 (Supplementary Data 11). Dataset3 contains all NCBI pathogen genomes submitted in the last 2 years, which was downloaded from https://www.ncbi.nlm.nih.gov/pathogens/microbigge/#[25] with "collection date" between 02/20/2019 to 02/10/2021 (Supplementary Data 12). Dataset2 covers a total of 362 taxa (42,481 genomes). The ARG sequences of the initial set were searched in three testing datasets using the same criteria described before ($e$-value of 1e−5, 90% aa similarity over 80% aa hit length). The ARG sequences of the initial set were mapped to all ARG sequences in AMRfinderPlus[25] by the same cutoff.

**Reporting summary**. Further information on research design is available in the Nature Research Reporting Summary linked to this article.

## Data availability

Details of methods, data, and scripts are all available in the Supplementary Information. We developed a bioinformatic tool arg_ranker v2.0[51] (https://github.com/caozhichongchong/arg_ranker and https://doi.org/10.5281/zenodo.5112502) for detecting ARGs and assessing the ARG risks in metagenomes and genomes (details in Supplementary Methods). Source data and processed data generated in this study are publicly available online (https://doi.org/10.6084/m9.figshare.15001053). The sequences of Rank I-II ARGs are available in Supplementary Data 2 and 3, and ranking information of all ARGs of the initial set is available in Supplementary Data 4. The prevalence of ARGs in FMT datasets (human gut microbiome genomes and metagenomes) and clinical datasets used for validation are available in Supplementary Data 6–12. The presence and abundance of ARGs of the initial set in 54,718 NCBI bacterial genomes (downloaded on 02/19/2019), 15,738 (all available after quality screening) NCBI plasmids (downloaded on 02/19/2019), and 854 global metagenomes of Illumina shotgun sequencing (downloaded on 02/19/2019) were downloaded from ARG-OSP (https://args-osp.herokuapp.com/). The validation datasets collected after 02/19/2019 were downloaded via (1) https://t.co/bdZxADGM7z (chromosomal genomes and plasmids from opportunistic pathogen isolates cultured from a tertiary hospital). (2) https://www.ebi.ac.uk/ena/browser/view/PRJEB31632 (116 metagenomes collected from the same tertiary hospital). (3) https://www.ncbi.nlm.nih.gov/pathogens/microbigge/# (all NCBI pathogen genomes submitted in the last 2 years). The databases of mobile genetic element were available online: the integron database (Additional file 1 from https://microbiomejournal.biomedcentral.com/articles/10.1186/s40168-018-0516-2#Sec15) and immeDB database (http://immedb.gutfun.org/).

## Code availability

All code is available on arg_ranker v2.0[51] (https://github.com/caozhichongchong/arg_ranker and https://doi.org/10.5281/zenodo.5112502) for detecting ARGs and assessing the ARG risks in metagenomes and genomes.

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

## Acknowledgements

Dr. A.N. Zhang, Dr. L.G. Li, and X. Yin acknowledge the University of Hong Kong for the postgraduate studentships. S.Z., and C.D. acknowledge the MIT for the postgraduate studentships. The authors would like to thank the Hong Kong Theme Based Research (T21-705/20-N), Broad Institute (Broad Next 10 grant 4000017), and Center for Microbiome Informatics and Therapeutics at MIT for the financial support.

## Author contributions

The research topic was developed by A.N.Z., E.J.A, and T.Z., A.N.Z. developed the pipeline, analysed the data, and wrote the manuscript. J.M.G. contributed suggestions on data analysis, statistic tests, manuscript preparation, and revised the manuscript. C.D. and S.Z. contributed suggestions on data analysis and revised the manuscript. M.G. and M.P. processed and provided gut microbiome genomic data. X.Y. provided suggestions in ARG analysis and L.L. contributed suggestions in training datasets collection (metagenomes and genomes). M.C.M. van Loosdrecht, M.R.G., W.P.H., J.M.T., and E.T. advised on the pipeline development and revised this manuscript. K.M. contributed suggestions on the logic flow of the manuscript and revised the manuscript. E.J.A. and T.Z. guided the pipeline development, data analysis, and revised this manuscript.

## Competing interests

The authors declare no competing interests.
