## [Peer Review File · Nature Communications]

REVIEWER COMMENTS

Reviewer #1 (Remarks to the Author):

Zhang and colleagues address an important problem associated with live microbial therapeutics, which is potential transfer of antibiotic resistance genes (ARGs). Given the ubiquity and wide range of ARGs, they propose a framework that ranks their potential risk based on their enrichment in human-associated environments, gene mobility, and microbial host pathogenic potential. Using this framework the authors have identified 35 of currently WHO-identified highest risk ARGs and an additional 38 ARGs in this category. They validated the framework using three clinical datasets, including two of pathogen-associated genomes and plasmids, and one of tertiary hospital metagenomes. In addition, they examined metagenomics of healthy stool donors. The authors conclude that (1) virtually no human donor-based microbiota product is completely safe and (2) their framework can be used in development of synthetic consortia of microbial therapeutics.

The investigators should be congratulated on tackling a complex and important problem, and trying to introduce a potentially useful framework to address it. However, I do find one of their conclusions to be premature and potentially dangerous to public health.

Until there is a viable alternative to donor-based microbiota products to treat very sick patients experiencing very high burdens of antibiotic exposure, e.g., patients with recurrent *C. difficile* infections, it is very premature to dismiss this highly effective solution as early as the second sentence in the abstract. Of course, FMT products have risks and unfortunate outcomes have been well documented. However, the authors fail to also acknowledge that in general FMT has been associated with an overall decrease in the burden of ARGs in patients with *C. difficile* infections as well as other diseases, e.g., advanced liver disease. The fact is that sick patients don't live in germ-free bubbles. In fact, they live in extremely high-risk human associated environments and vulnerable to colonization by a multitude of pathobionts in their immediate surroundings. While a defined consortium of ARG-free microbes being administered into a patient (which is yet to be developed into any advanced Phase clinical trial) may at first glance appear to be safer, the real measure of safety should be the final microbial community in the patient. That microbial community will be made of original microbiota, administered microbiota, and microbes from the environment!

The authors could have interrogated some published databases of FMT recipients, e.g., *C. difficile* infections, liver disease to test how well the framework performed with this treatment. In absence of that, they should adjust their writing and allow for some grey, which is intrinsic to all medicine. Every decision in medicine is an evaluation of risk and benefit. It is real life - there is no black and white. The framework used by the authors may still be used to stratify donors by relative risk of ARG burden. Even more importantly, the proposed framework can be used in clinical trial design to see whether any microbiota-based intervention results in greater or lesser burden of high risk ARGs.

Reviewer #2 (Remarks to the Author):

In the work entitled "The Framework for Assessing The Health Risk of Antimicrobial Resistance in Microbiome-Based Therapeutics" The authors performed an extensive computational analysis with the aim of proposing a framework capable of identifying ARGs in samples for application in microbiome-based therapy, such as fecal transplantation. Considering the distribution of ARGs in the ESKAPE group, the authors created a threat rank to classify potentially harmful samples to their receptors, considering the presence of ARGs. The work brings an interesting methodology applied to a matter of

great relevance.

After completing the reading, I got an impression that the work deals with two issues, correlates, but different. In the first part of the work the authors made a meta-analysis of a database previously developed and published by the Group (Xiaole, 2018; Zhang, 2020), presenting ARG distribution statistics, which does not correspond to the title of the paper. In the second part of the work there is, indeed, a focus on developing a framework for ARGs analysis in microbioma samples for therapeutic use, consistent with the title of the article. At the discussion session, the first part of the work was practically not quoted.

My points here:

- 1- What is the novelty presented in the first part of the work in relation to the others already published?
- 2- How does the first part relate to the second?

An Ni Zhang, Chen-Ju Hou, Mishty Negi, Li-Guan Li, Tong Zhang, Online searching platform for the antibiotic resistome in bacterial tree of life and global habitats, *FEMS Microbiology Ecology*, Volume 96, Issue 7, July 2020, fiae107,

Xiaole Yin, Xiao-Tao Jiang, Benli Chai, Liguan Li, Ying Yang, James R Cole, James M Tiedje, Tong Zhang, ARGs-OAP v2.0 with an expanded SARG database and Hidden Markov Models for enhancement characterization and quantification of antibiotic resistance genes in environmental metagenomes, *Bioinformatics*, Volume 34, Issue 13, 01 July 2018, Pages 2263–2270

The possibility of methodologies such as fecal transplantation of healthy patients to carry potentially harmful organisms as well as ARGs has been considered for some time. However, so far, the FMT brings more benefits than damage. The reported cases are few, as well as death cases provenly caused by bacteria originating from a healthy donor. Even if the transfer of a pathogen containing a risk I ARGs is not necessarily true that this gene will be distributed and expressed throughout the bacterial community. That said, I suggest the authors to decrease the focus on the "threat" of the FMT and focus on the development of the framework.

Line 146-152: This session describes the presence of total antibiotics in different environments. It is difficult to understand what "Total Environmental Concentration of Antibiotics" means. Looking in some of the references cited, both in the manuscript and in the supplementary material (Table S3), I did not find any measure of antibiotic concentration, nor what classes were found. The very determination of antibiotic concentration in an environmental sample has its challenges. It would be very enlightening if the authors provided accurate information on obtaining the environmental concentration data of antibiotics.

Line 173-177: Include a citation that the analyzed genomes, plasmids and metagenomes are part of the ArgS-OSP database. It would also be enlightening to bring information about the diversity of the dataset used. In the ArgS-OSP site it is possible to verify that the genomes database is formed majority by opportunistic pathogenic bacteria. It would be interesting to see this same analysis in a group of control genomes. Choi et al created a reference database for genomes of soil bacteria that may be useful for this analysis. Youngfei et al (2013) performed a prospecting by resistance genes in fecal microbiomes of 162 individuals. In addition to variance between populations, they verified that tetracycline resistance genes are the most dispersed, while betalactamase coding genes were rare.

Choi, J., Yang, F., Stepanauskas, R. et al. Strategies to improve reference databases for soil microbiomes. *ISME J* 11,829–834 (2017).

Hu, Y., Yang, X., Qin, J. et al. Metagenome-wide analysis of antibiotic resistance genes in a large cohort of human gut microbiota. *Nat Commun* 4, 2151 (2013).

Lines 324-331: Considering the analysis of healthy donor microbiome, it is possible to conclude that the bacteria of the ESKAPE group are not the most represented (Figure S7). The Gammaproteobacteria class, which contributes to the ESKAPE group with the Enterobacteriaceae family and the species *Klebsiella pneumoniae*, *Pseudomonas aeruginosa* and *Acinetobacter baumannii* represent a fraction of the total. Nor *Staphylococcus* or *Enterococcus* genres were found. What is the relevance of using the ESKAPE group as a pathogenic standard? It seems that everything comes down to gammaproteobacterias. In addition, it would be interesting to see a clustering analysis between ARGs risk I of healthy donors X ARGs risk I of microbioma of patients. If the same ARGs are found, the donor microbiota would be, indeed, a risk factor?

Lines 382-396: D'Costa et al (2011) reported the identification of several antibiotic resistance genes in frozen soils of permafrost of 30 thousand years. Particularly, they have shown that a 30,000-year-old *vanA* gene is able to confer vancomycin resistance in current bacteria, showing that this gene was already widely distributed thousands of years before the emergence of industrialized antibiotics.

'Costa, V., King, C., Kalan, L. et al. Antibiotic resistance is ancient. *Nature* 477, 457–461 (2011).
<https://doi.org/10.1038/nature10388>

Lines 420-422: antibiotic resistance is a phenomenon, in many ways, little understood. The study in question adds more knowledge about the dispersion of resistance genes in microbiome samples for therapy, but I think there is much to be done before we reach a "rigid, clear and convenient" recommendation. It is necessary to consider that the computational prediction of ARGs does not always reflect the phenotype.

Lines 443-446: In addition to the suggestion itself, which seems to me plausible in the near future, the authors should include a commentary on the difficulty in developing primers capable of amplifying, with a high degree of certainty, only ARGs of interest in a wide range of bacterial species. By this approach, how could it be demonstrated that an ARG classified as risk I-II by the proposed framework is also in a mobile element in the tested microbiome?

Minor

Line 42: "... Safe to include in the cocktail ..."

Line 72: The case listed in Ref 4 report one death.

Line 119-124: It's very confusing. Lots of information in a single sentence.

Line 366-369: "Future studies". The statement in this session would be better understood with the presentation of a real example of how an ARGs phylogenetic analysis could be transformed into a control strategy.

Line 512: 563 or 560 Metagenome datasets?

Reviewer #3 (Remarks to the Author):

Zhang et, al proposed a framework to assess the health risk of ARGs for microbial therapeutics considering the wide distribution of ARGs in the human gut and the rising needs of FMT as an effective treatment of various diseases such as C. Diff infection, IBD, immunotherapy of cancer. Without effective evaluation, there is a huge hidden risk of transferring drug-resistance bugs to the recipient. The framework is further validated with three datasets and demonstrates its effectiveness on enumerated the health risk of ARGs in hospital spots. This work gives the first try to fill the important gap of developing a usable method to evaluate the risk of FMT. In light of the tragic death of patients from FMT that happened in the USA, there is an urgent need for risk assessment of FMT donor samples.

Considering the importance and widespread interest of this topic, it would be great if the authors can supply a simplified way to apply the framework for other field researchers, for example, a script/tool that can easily be applied to the donor samples, may not in this release but in the future at least. For assessment of the safety of FMT donors, only ranking the resistome may not enough. While some donors do not have RANK I resistance genes, but they may have the ability to produce endotoxic chemicals. The addition of virulence factors assessment will help to strengthen the framework for safety FMT or selection of strains.

Overall, this is a complex work and urgently needed to be sorted out. Applying FMT to cure disease has a "super donor" phenomenon and the success of FMT may not simply due to the transferring of the bacteria, the other parts, for example, virome and even debris of dead bacteria, may also very important for the success of the microbial therapeutics. This should be discussed in the manuscript.

REVIEWER COMMENTS

Reviewer #1 (Remarks to the Author):

Zhang and colleagues address an important problem associated with live microbial therapeutics, which is potential transfer of antibiotic resistance genes (ARGs). Given the ubiquity and wide range of ARGs, they propose a framework that ranks their potential risk based on their enrichment in human-associated environments, gene mobility, and microbial host pathogenic potential. Using this framework the authors have identified 35 of currently WHO-identified highest risk ARGs and an additional 38 ARGs in this category. They validated the framework using three clinical datasets, including two of pathogen-associated genomes and plasmids, and one of tertiary hospital metagenomes. In addition, they examined metagenomics of healthy stool donors. The authors conclude that (1) virtually no human donor-based microbiota product is completely safe and (2) their framework can be used in development of synthetic consortia of microbial therapeutics.

The investigators should be congratulated on tackling a complex and important problem, and trying to introduce a potentially useful framework to address it. However, I do find one of their conclusions to be premature and potentially dangerous to public health.

Until there is a viable alternative to donor-based microbiota products to treat very sick patients experiencing very high burdens of antibiotic exposure, e.g., patients with recurrent *C. difficile* infections, it is very premature to dismiss this highly effective solution as early as the second sentence in the abstract. Of course, FMT products have risks and unfortunate outcomes have been well documented. However, the authors fail to also acknowledge that in general FMT has been associated with an overall decrease in the burden of ARGs in patients with *C. difficile* infections as well as other diseases, e.g., advanced liver disease. The fact is that sick patients don't live in germ-free bubbles. In fact, they live in extremely high-risk human associated environments and vulnerable to colonization by a multitude of pathobionts in their immediate surroundings. While a defined consortium of ARG-free microbes being administered into a patient (which is yet to be developed into any advanced Phase clinical trial) may at first glance

appear to be safer, the real measure of safety should be the final microbial community in the patient. That microbial community will be made of original microbiota, administered microbiota, and microbes from the environment!

The authors could have interrogated some published databases of FMT recipients, e.g., *C. difficile* infections, liver disease to test how well the framework performed with this treatment. In absence of that, they should adjust their writing and allow for some grey, which is intrinsic to all medicine. Every decision in medicine is an evaluation of risk and benefit. It is real life - there is no black and white. The framework used by the authors may still be used to stratify donors by relative risk of ARG burden. Even more importantly, the proposed framework can be used in clinical trial design to see whether any microbiota-based intervention results in greater or lesser burden of high risk ARGs.

Reply: Thank you so much for your positive comments and insightful suggestions!

Your comments have greatly improved our work.

It has made our story much more solid and clearer.

We have removed this perspective and rewritten the manuscript to focus mainly on the development of the risk framework.

We have now acknowledged that “FMT has been proven a promising treatment for some dysbioses of the human gut microbiome, e.g. C. difficile infection, inflammatory bowel disease (IBD), and Hematopoietic Cell Transplantation (HCT)” (Lines 416-419).

In the discussion, we talked about some potential application of this framework: 1) “to control the transmission of high-risk AMR to humans by screening out high-risk strains for microbiome-based therapeutic and live biotherapeutic products” (Lines 364-366); 2) “this framework can be used in clinical trial design to investigate which microbiota-based intervention results in the least burden of high risk ARGs” (Lines 423-425); 3) “Applying the risk framework to the metagenomes of FMT donors would potentially improve the safety of FMT by screening for FMT donors with a low abundance of Rank I-II ARGs” (Lines 421-422).

We also pointed out that “only considering high-risk resistome is not sufficient for the assessment of the safety of FMT donors. Further investigation into the virulence factors,

endotoxic chemicals, virome composition, and even debris of dead bacteria will help strengthen the guidance for the safety of FMT and other microbiome-based therapeutics” (Lines 430-434). It’s a great point that we can “use this framework in clinical trial design to see whether any microbiota-based intervention results in greater or lesser burden of high risk ARGs”, we have included now in the discussion (Lines 430-434).

We really appreciate the comments of you to help improve our work.

Reviewer #2 (Remarks to the Author):

In the work entitled "The Framework for Assessing The Health Risk of Antimicrobial Resistance in Microbiome-Based Therapeutics" The authors performed an extensive computational analysis with the aim of proposing a framework capable of identifying ARGs in samples for application in microbiome-based therapy, such as fecal transplantation. Considering the distribution of ARGs in the ESKAPE group, the authors created a threat rank to classify potentially harmful samples to their receptors, considering the presence of ARGs. The work brings an interesting methodology applied to a matter of great relevance.

After completing the reading, I got an impression that the work deals with two issues, correlates, but different. In the first part of the work the authors made a meta-analysis of a database previously developed and published by the Group (Xiaole, 2018; Zhang, 2020), presenting ARG distribution statistics, which does not correspond to the title of the paper. In the second part of the work there is, indeed, a focus on developing a framework for ARGs analysis in microbioma samples for therapeutic use, consistent with the title of the article. At the discussion session, the first part of the work was practically not quoted.

My points here:

- 1- What is the novelty presented in the first part of the work in relation to the others already published?
- 2- How does the first part relate to the second?

An Ni Zhang, Chen-Ju Hou, Mishty Negi, Li-Guan Li, Tong Zhang, Online searching platform for the antibiotic resistome in bacterial tree of life and global habitats, FEMS Microbiology

Ecology, Volume 96, Issue 7, July 2020, fiaa107,

Xiaole Yin, Xiao-Tao Jiang, Benli Chai, Liguan Li, Ying Yang, James R Cole, James M Tiedje, Tong Zhang, ARGs-OAP v2.0 with an expanded SARG database and Hidden Markov Models for enhancement characterization and quantification of antibiotic resistance genes in environmental metagenomes, *Bioinformatics*, Volume 34, Issue 13, 01 July 2018, Pages 2263–2270

Reply: Thank you so much for your positive comments and helpful suggestions!

We completely agree with you that the first part of our work was less relevant to the title.

We have now removed the part of ARG distribution statistics from the MS.

We have rewritten the first part of the MS to focus on the characteristics of each risk Rank and the validation of the risk framework by WHO-listed ARGs and recent datasets.

It made our story much clearer.

Your comments have greatly improved our work. Thank you very much!

The possibility of methodologies such as fecal transplantation of healthy patients to carry potentially harmful organisms as well as ARGs has been considered for some time. However, so far, the FMT brings more benefits than damage. The reported cases are few, as well as death cases provenly caused by bacteria originating from a healthy donor. Even if the transfer of a pathogen containing a risk I ARGs is not necessarily true that this gene will be distributed and expressed throughout the bacterial community. That said, I suggest the authors to decrease the focus on the "threat" of the FMT and focus on the development of the framework.

Reply: Thank you so much for your comments!

We have rewritten the paper and focused mainly on the development of the risk framework (refer to replies to Reviewer #1).

It has made our story much more solid and clearer. We really appreciate the comments of you to help improve our work.

Line 146-152: This session describes the presence of total antibiotics in different environments.

It is difficult to understand what "Total Environmental Concentration of Antibiotics means. Looking in some of the references cited, both in the manuscript and in the supplementary material (Table S3), I did not find any measure of antibiotic concentration, nor what classes were found. The very determination of antibiotic concentration in an environmental sample has its challenges. It would be very enlightening if the authors provided accurate information on obtaining the environmental concentration data of antibiotics.

Reply: Thank you for your comments!

We have now clarified the method in Supplementary Method and provided the raw data of the concentrations of antibiotics and breakdown of calculations in Table S1.

“We searched for literatures that surveyed the antibiotics of classes quinolones, sulfonamides, macrolides, tetracycline, and chloramphenicol. Raw data of the concentrations of antibiotics was collected from the tables and/or supplementary data from the original literatures (Table S1). The concentrations of all antibiotics of a sample were summed up to represent the total concentration of antibiotics (breakdown of calculations in Table S1 details). The total concentration of antibiotics was further normalized to the unit of ng/g (for solid samples) or ng/L (for liquid samples)”.

Line 173-177: Include a citation that the analyzed genomes, plasmids and metagenomes are part of the Args-OSP database. It would also be enlightening to bring information about the diversity of the dataset used. In the Args-OSP site it is possible to verify that the genomes database is formed majority by opportunistic pathogenic bacteria. It would be interesting to see this same analysis in a group of control genomes. Choi et al created a reference database for genomes of soil bacteria that may be useful for this analysis. Youngfei et al (2013) performed a prospecting by resistance genes in fecal microbiomes of 162 individuals. In addition to variance between populations, they verified that tetracycline resistance genes are the most dispersed, while betalactamase coding genes were rare.

Choi, J., Yang, F., Stepanauskas, R. et al. Strategies to improve reference databases for soil microbiomes. ISME J 11,829–834 (2017).

Hu, Y., Yang, X., Qin, J. et al. Metagenome-wide analysis of antibiotic resistance genes in a large cohort of human gut microbiota. Nat Commun 4, 2151 (2013).

Reply: Thank you for your comments and for sharing those papers with us!

We have clarified that the analyzed genomes, plasmids and metagenomes were part of ARGs-OSP database and included the reference in the manuscript (Lines 491-497).

It's a great point to investigate the risk of ARGs in genomes from different environments. We have added this point in the discussion that "this framework could be applied to prioritize bacterial strains carrying ARGs with high risk of environmental dissemination and identify potential environmental hotspots by investigating sequencing data" (Lines 443-446).

In the framework, we used the environmental metagenomes (non-anthropogenically impacted environments) as the control group to the metagenomes of anthropogenically impacted environments. By comparing the abundance of ARGs in these two groups of metagenomes, we identified ARGs that were significantly "human-associated" (first factor of the framework). Genomes from ARGs-OSP were used to check whether those ARGs were present in known pathogens (third factor of the framework). Thus, we focused on genomes of opportunistic pathogenic bacteria.

*As to fecal microbiome, we analyzed ARGs in 1,921 representative fecal microbiome genomes isolated from 59 healthy donors (489 strains from FMT donors and 1,432 strains from non-FMT global donors). Though we didn't discuss in this study the prevalence of individual resistance gene, we found that tetracycline resistance gene *tetQ* and betalactamase *CfxA2* were both highly prevalent in human gut microbiome genomes (top five), and multidrug resistance genes were the most prevalent in general.*

Lines 324-331: Considering the analysis of healthy donor microbiome, it is possible to conclude that the bacteria of the ESKAPE group are not the most represented (Figure S7). The Gammaproteobacteria class, which contributes to the ESKAPE group with the Enterobacteriaceae family and the species *Klebsiella pneumoniae*, *Pseudomonas aeruginosa* and *Acinetobacter baumannii* represent a fraction of the total. Nor *Staphylococcus* or *Enterococcus* genres were found. What is the relevance of using the ESKAPE group as a pathogenic standard? It seems that everything comes down to gammaproteobacterias.

In addition, it would be interesting to see a clustering analysis between ARGs risk I of healthy donors X ARGs risk I of microbioma of patients. If the same ARGs are found, the donor microbiota would be, indeed, a risk factor?

Reply: Thank you for your comments!

We agree that ESKAPE groups were not the most represented in healthy donor microbiome. We first classified human gut microbiome genomes/strains into pathogenic and non-pathogenic strains because Rank I ARGs, by definition, would be highly prevalent in pathogens (ESKAPE and multidrug resistant Escherichia coli). We observed that high-risk ARGs were significantly enriched in pathogenic strains ($p = 2E-88$ by fisher exact test) compared to non-pathogenic strains (Lines 397-399). Thus, we recommend excluding the entire set of pathogenic strains and the subset of non-pathogenic strains with high-risk ARGs from microbiome-based therapeutic (Lines 399-402).

Although it's true that our dataset didn't contain any Staphylococcus (Enterococcus strains were labeled as purple bars in Figure S7), we would still recommend excluding Staphylococcus from microbiome-based therapeutic.

In this study, we only focused on ESKAPE pathogens and multidrug resistant Escherichia coli. We pointed out this limitation in the discussion and future work would also benefit from comprehensive pathogen lists, and manual curation is highly recommended to differentiate commensals and opportunistic pathogens (i.e. Bacteroides and Lactobacillus species) (Lines 473-476).

It's very good point to see whether and what high-risk ARGs are transferred through FMT. As we have now rewritten the paper to focus mainly on the risk framework development. We only discussed FMT was one of the potential applications in the future. We didn't investigate into the FMT metagenomes.

We have mentioned this point in the discussion that "future studies should study the efficiency of transferring high-risk ARGs through FMT using the metagenomes from FMT donors and recipients" (Lines 426-429).

Lines 382-396: D'Costa et al (2011) reported the identification of several antibiotic resistance

genes in frozen soils of permafrost of 30 thousand years. Particularly, they have shown that a 30,000-year-old vanA gene is able to confer vancomycin resistance in current bacteria, showing that this gene was already widely distributed thousands of years before the emergence of industrialized antibiotics.

Costa, V., King, C., Kalan, L. et al. Antibiotic resistance is ancient. *Nature* 477, 457–461 (2011). <https://doi.org/10.1038/nature10388>

Reply: Thank you for your comments and for sharing this paper!

It's a very good point. We have now cited the paper and discussed about it in the manuscript: "it indicates that vanA in human microbiome could originate from natural microbes thousands of years before the emergence of industrialized antibiotics" (Lines 330-333). "Thus, sulI and vanA, which our framework did not identify as high-risk ARGs, represented ARGs that have been widespread for a long time" (Lines 339-341).

Lines 420-422: antibiotic resistance is a phenomenon, in many ways, little understood. The study in question adds more knowledge about the dispersion of resistance genes in microbiome samples for therapy, but I think there is much to be done before we reach a "rigid, clear and convenient" recommendation. It is necessary to consider that the computational prediction of ARGs does not always reflect the phenotype.

Reply: Thank you for your comments!

We completely agree with you.

We have removed the statement of "rigid, clear and convenient recommendation" and pointed it out in the discussion that "only considering high-risk resistome is not sufficient for the assessment of the safety of FMT donors. Further investigation into the virulence factors, endotoxic chemicals, virome composition, and even debris of dead bacteria will help strengthen the guidance for the safety of FMT and other microbiome-based therapeutics" (Lines 430-434). We also clarified that "the computational prediction of ARGs does not always reflect the phenotypes" (Lines 425-426) and "future studies could benefit from characterizing the

phenotypes of ARGs in primary human pathogens and human gut commensals, and report clinical evidence” (Lines 466-468).

It has made our conclusions much more solid.

Lines 443-446: In addition to the suggestion itself, which seems to me plausible in the near future, the authors should include a commentary on the difficulty in developing primers capable of amplifying, with a high degree of certainty, only ARGs of interest in a wide range of bacterial species. By this approach, how could it be demonstrated that an ARG classified as risk I-II by the proposed framework is also in a mobile element in the tested microbiome?

Reply: *Thank you for your comments.*

It's a very good point.

We have discussed about the difficulty of designing ARG primers that “one critical factor to consider when screening for ideal primers is the capability of amplifying, with a high degree of specificity, only ARGs of interest in a wide range of bacterial species” (Lines 453-456).

We completely agreed that only using ARG primers does not only target those on mobile elements and we have removed the original statement. We have now clarified that “applying ARG primers to the plasmidome and combining ARG primers with primers targeting MGEs (e.g. integrons) via epicPCR would be useful to target mobile high-risk ARGs” (Lines 456-458).

Minor

Line 42: "... Safe to include in the cocktail ..."

Reply: *Thank you for your comments. We have revised accordingly.*

Line 72: The case listed in Ref 4 report one death.

Reply: *Thank you for your comments. We have now removed this sentence.*

Line 119-124: It's very confusing. Lots of information in a single sentence.

Reply: Thank you for your comments.

We have revised it to “The first metric our framework considers reflects the fact that ARGs that are much more abundant in anthropogenically impacted environments than in non-impacted environments are most likely either to be associated with human or livestock microbiomes, or to be directly selected for resistance to clinical or livestock antibiotics, or both. We refer to this as the enrichment of putative ARGs in “human-associated” environments” (Lines 101-106).

Line 366-369: "Future studies". The statement in this session would be better understood with the presentation of a real example of how an ARGs phylogenetic analysis could be transformed into a control strategy.

Reply: Thank you for your comments.

We have now presented an example that “ARG phylogenetic analysis can help differentiate horizontally acquired resistance from mutation-based resistance, which evolves differently and therefore requires different strategies to control (the former one would more likely to be driven by anthropogenic pollution with selective agents and microbes of human or domestic animal origin; and the latter would more likely to be selected by antibiotic treatment inside human or domestic animal body)” (Lines 308-314).

Line 512: 563 or 560 Metagenome datasets?

Reply: Thank you for your comments. We have corrected the inconsistency.

Reviewer #3 (Remarks to the Author):

Zhang et, al proposed a framework to assess the health risk of ARGs for microbial therapeutics considering the wide distribution of ARGs in the human gut and the rising needs of FMT as an effective treatment of various diseases such as C. Diff infection, IBD, immunotherapy of cancer. Without effective evaluation, there is a huge hidden risk of transferring drug-resistance bugs to the recipient. The framework is further validated with three datasets and demonstrates its

effectiveness on enumerated the health risk of ARGs in hospital spots. This work gives the first try to fill the important gap of developing a usable method to evaluate the risk of FMT. In light of the tragic death of patients from FMT that happened in the USA, there is an urgent need for risk assessment of FMT donor samples.

Considering the importance and widespread interest of this topic, it would be great if the authors can supply a simplified way to apply the framework for other field researchers, for example, a script/tool that can easily be applied to the donor samples, may not in this release but in the future at least.

Reply: Thank you so much for your positive comments and helpful suggestions!

It's a great point and has greatly improve our study.

We have now designed a bioinformatic tool for detecting ARGs and assessing the ARG risks in metagenomes and genomes (https://github.com/caozhichongchong/arg_ranker) (Lines 481-484).

For assessment of the safety of FMT donors, only ranking the resistome may not enough. While some donors do not have RANK I resistance genes, but they may have the ability to produce endotoxic chemicals. The addition of virulence factors assessment will help to strengthen the framework for safety FMT or selection of strains.

Overall, this is a complex work and urgently needed to be sorted out. Applying FMT to cure disease has a “super donor” phenomenon and the success of FMT may not simply due to the transferring of the bacteria, the other parts, for example, virome and even debris of dead bacteria, may also very important for the success of the microbial therapeutics. This should be discussed in the manuscript.

Reply: Thank you so much for your positive comments and helpful suggestions!

We completely agree that only considering high-risk resistome is not enough and the factors mentioned in the comments are critical for the success of FMT.

We have now discussed about it that “only considering high-risk resistome is not sufficient for the assessment of the safety of FMT donors. Further investigation into the virulence factors,

endotoxic chemicals, virome composition, and even debris of dead bacteria will help strengthen the guidance for the safety of FMT and other microbiome-based therapeutics” (Lines 430-434).

It has made our conclusions much more solid.

We really appreciate the comments of you to help improve our work.

REVIEWER COMMENTS

Reviewer #1 (Remarks to the Author):

Zhang and colleagues have improved their manuscript. However, its prescriptions to “world-wide health authorities” based on this work are not especially helpful.

1. While the authors cite “WHO and previous literature” with respect to “clinically relevant ARGs”, the listing of the major families of Rank I ARGs does not conform with clinical reality. Antibiotics in their Figure 3 that are most clinically relevant because of their usage are beta-lactams, quinolones, and vancomycin. Tetracycline, trimethoprim, chloramphenicol, aminoglycosides are of little clinical interest either because of their impotence against difficult pathogens or prohibitive side effect profiles.
2. The authors suggest removing all ‘non-pathogenic’ strains carrying high risk ARGs from all microbiome-based therapeutics. At the same time, they recognize the important roles of non-pathogenic members of the microbiome in colonization resistance and overall functionality of the gut microbiota. That is essentially a prescription for requiring assembly of entirely synthetic microbial consortia nearing the complexity of natural microbiota. This is hardly a practical suggestion. Besides, we are not given a clear definition of ‘pathogenicity’. Any microbe that finds itself in a wrong compartment in the body can be pathogenic.
3. While the authors removed their FMT as a threat to public health language, their message for world-wide health authorities for donor exclusion criteria is equivalent to a recommendation of banning FMTs. This is not justified until we have clinical trials supporting alternative live biotherapeutic agents to be superior. Again, I would suggest that the burden of ARGs after treatment can be used as a helpful metric in such evaluations. I think an attempt to create a meaningful schema for selecting ARG targets in the patient’s microbiome can be meaningful, while application in manufacturing is premature.
4. The FDA currently does require testing donor FMT material for ESBL, VRE, MRSA, CRE. Were the donor samples used in this study obtained before or after this guidance? What fraction of donor samples passed these established tests done in clinical, CLIA-certified laboratories?
5. Would any donors pass the author’s threshold of ARG tolerance? What can we learn about that super-donor population?
6. Do the investigators recommend the FDA replace their current requirements for MDRO testing with their yet-to-be-developed PCR panel? Shouldn’t there be some intermediate steps for validation?

In conclusion, I find the effort of sorting through the ARG complexity admirable and important. However, I think the ranking scheme also needs a clinical filter for relevancy and the paper needs some reality checks.

Reviewer #2 (Remarks to the Author):

The revised version of the manuscript showed an improvement over the previous version. All suggestions were considered and discussed. As a final comment, I leave a suggestion for the title “A Omics-based Framework for Assessing the Health Risk of Antimicrobial Resistance”

Reviewer #3 (Remarks to the Author):

The authors solved my questions well. An easy-to-use pipeline is deposited by the authors on Github to make other researchers easier to use this framework.

I have no further comments on the manuscript.

REVIEWER COMMENTS

Reviewer #1 (Remarks to the Author):

Zhang and colleagues have improved their manuscript. However, its prescriptions to “world-wide health authorities” based on this work are not especially helpful.

Re: Thank you so much for your helpful comments!

What we propose in this study is an idea for reducing ARG resistance risk which can be applied to FMT regulation by incorporating our risk framework with MDRO testing. We hope that our framework can, in the future, help to rank the FMT donors by their abundance of high-risk ARGs in gut microbiome metagenomes and prioritize FMT donors having lower ARG abundances. This idea still requires more validation before being implemented as a part of the prescriptions. We have now rewritten the discussion part in the manuscript and also revised the abstract (Lines 51-53, 428-445).

We really appreciate the comments of the reviewer. These comments help us rethink what we proposed and clarify our point in a clearer way in the revised manuscript.

1. While the authors cite “WHO and previous literature” with respect to “clinically relevant ARGs”, the listing of the major families of Rank I ARGs does not conform with clinical reality. Antibiotics in their Figure 3 that are most clinically relevant because of their usage are beta-lactams, quinolones, and vancomycin. Tetracycline, trimethoprim, chloramphenicol, aminoglycosides are of little clinical interest either because of their impotence against difficult pathogens or prohibitive side effect profiles.

Re: Thank you for your helpful comments!

In our study, we define “clinically relevant ARGs” as ARGs that were reported to have caused antibiotic treatment failure and/or outbreak in hospitals. We collected a list of the “clinically relevant ARGs” that were prioritized by the WHO report because they rendered many antibiotic agents ineffective; or reported by literatures to have caused treatment failure and/or outbreak in hospitals across the world (reference 25-34 in supplementary information); or reported to be widespread on MGEs in clinically relevant environments such as hospital wastewater (reference 25,35-38 in the Supplementary Information). We have now clarified the definition and literature review in both the main text of our manuscript (Lines 210-214) and the Supplementary Information.

Note that according to the 2016-2018 WHO Report on Surveillance of Antibiotic Consumption, tetracycline and trimethoprim are widely used in many countries in Europe, the Americas, and

Africa (10%-20% of total consumption of human antimicrobial use). This report also indicates that aminoglycosides are widely used in many countries (~1% of total consumption).

2. The authors suggest removing all ‘non-pathogenic’ strains carrying high risk ARGs from all microbiome-based therapeutics. At the same time, they recognize the important roles of non-pathogenic members of the microbiome in colonization resistance and overall functionality of the gut microbiota. That is essentially a prescription for requiring assembly of entirely synthetic microbial consortia nearing the complexity of natural microbiota. This is hardly a practical suggestion. Besides, we are not given a clear definition of ‘pathogenicity’. Any microbe that finds itself in a wrong compartment in the body can be pathogenic.

Re: Thank you for your helpful comments!

In this study, we propose that this risk framework could help improve the safety of microbiome-based interventions (FMT and microbiome-based therapeutics) by facilitating the evaluation of the risk of various approaches and selecting the least risky among feasible approaches.

For FMT, we hope that our framework can be useful in the future to rank the FMT donors by their abundance of high-risk ARGs in gut microbiome metagenomes and prioritize FMT donors with lower ARG abundances. For microbiome-based therapeutics, we propose that our framework can help screen out high-risk strains. We found that non-pathogenic strains that carried a Rank I-II ARG varied from donor to donor so that we can pick up “clean” strains from different donors to design a complex assembly. This idea requires more validation in the future before implementation. We have now clarified it in the manuscript (Lines 410-414,432-445, 466-473).

How to design and implement microbial consortia nearing the complexity of natural microbiota is not the focus of this study. Also, we are not suggesting assembly of entirely synthetic microbial consortia nearing the complexity of natural microbiota. Sorry for not making this clear in the original manuscript. We now have clarified this point in the revised manuscript.

*We totally agree that ‘pathogenicity’ is not clear defined in the original manuscript. Here we consider pathogens to be those that exhibit multidrug resistance and virulence listed by the FDA; for example, *Clostridium difficile*, *Staphylococcus aureus*, *Klebsiella pneumoniae*, *Acinetobacter baumannii*, and *Pseudomonas aeruginosa*. We made this clear now in the revised manuscript. Thanks for the comment.*

3. While the authors removed their FMT as a threat to public health language, their message for world-wide health authorities for donor exclusion criteria is equivalent to a recommendation of banning FMTs. This is not justified until we have clinical trials supporting alternative live biotherapeutic agents to be superior. Again, I would suggest that the burden of ARGs after

treatment can be used as a helpful metric in such evaluations. I think an attempt to create a meaningful schema for selecting ARG targets in the patient's microbiome can be meaningful, while application in manufacturing is premature.

Re: Thank you for your helpful comments!

First of all, we don't want to ban FMT or restrict the number of FMT donors. Actually, we consider FMT a life-saving treatment with no alternative currently available in other ways. We should have made it clearer in the manuscript (Lines 422-426). Sorry for not being clear on this point in our original manuscript. We have revised our manuscript accordingly.

What we propose in this study is an idea for supplementing existing FMT regulation by incorporating our risk framework to evaluate the relative risks of individual samples and facilitate selecting the samples having least risk of introduction of antibiotic resistance. This idea still requires more validation in the future before implementation. We have now rewritten the discussion part in the manuscript (Lines 428-445).

It's a very good point that "the burden of ARGs after treatment can be used as a helpful metric in such evaluations" and we have discussed it in the manuscript that "this framework can be used in the clinical trial design to investigate which microbiota-based intervention results in the least burden of high risk ARGs" (Lines 437-439).

4. The FDA currently does require testing donor FMT material for ESBL, VRE, MRSA, CRE. Were the donor samples used in this study obtained before or after this guidance? What fraction of donor samples passed these established tests done in clinical, CLIA-certified laboratories?

Re: Thank you for your helpful comments!

The donor samples were obtained before this guidance from Broad Institute-OpenBiome Microbiome Library (BIO-ML) but all donors from OpenBiome were screened for MDRO, including ESBL-producing organisms, CRE, MRSA, and VRE.

[<https://www.openbiome.org/press-releases/2019/6/14/statement-on-fda-safety-alert>, <https://www.openbiome.org/safety>]. We have now clarified it in the manuscript (Lines 373-378).

5. Would any donors pass the author's threshold of ARG tolerance? What can we learn about that super-donor population?

Re: Thank you for your helpful comments!

We propose an idea to use our framework to rank the FMT donors by their abundance of high-risk ARGs in gut microbiome metagenomes. Instead of using a threshold of ARG tolerance, we hope that our framework can help prioritize FMT donors with a lower ARG burden. What we

propose in this study is an idea or suggestion for improving FMT regulation and this idea requires more validation in the future before implementation. We have now clarified it in the manuscript (Lines 432-437).

6. Do the investigators recommend the FDA replace their current requirements for MDRO testing with their yet-to-be-developed PCR panel? Shouldn't there be some intermediate steps for validation?

Re: Thank you for your helpful comments!

The risk framework proposed in our study cannot replace MDRO testing. The requirement of MDRO testing from FDA can strictly regulate the antimicrobial resistant pathogens in FMT. We would like to propose an idea to incorporating the risk framework with MDRO testing, which would help diminish the risk of novel resistance emerging from non-pathogens. This idea definitely requires many steps of validation in the future before implementation. We have now rewritten the discussion part in the manuscript (Lines 428-437).

In conclusion, I find the effort of sorting through the ARG complexity admirable and important. However, I think the ranking scheme also needs a clinical filter for relevancy and the paper needs some reality checks.

Re: Thank you for your helpful comments!

Your comments, especially how to incorporate with current FDA guidelines, have greatly improved our work. We really appreciate your comments!

Reviewer #2 (Remarks to the Author)

The revised version of the manuscript showed an improvement over the previous version. All suggestions were considered and discussed. As a final comment, I leave a suggestion for the title "A Omics-based Framework for Assessing the Health Risk of Antimicrobial Resistance"

Re: Thank you for your helpful comment!

We have revised the title as you suggested. All your previous comments have greatly improved our work. We really appreciate your comments!

Reviewer #3 (Remarks to the Author):

The authors solved my questions well. An easy-to-use pipeline is deposited by the authors on Github to make other researchers easier to use this framework.

I have no further comments on the manuscript.

Re: Thank you for your time and efforts handling our manuscript!

Your previous comments, especially the idea of developing a bioinformatic pipeline, have greatly improved our work. We really appreciate your comments!